# Can Digital Transformation Promote the Rapid Recovery of Cities from the COVID-19 Epidemic? An Empirical Analysis from Chinese Cities

**DOI:** 10.3390/ijerph19063567

**Published:** 2022-03-17

**Authors:** Jiaojiao Liu, Shuai Liu, Xiaolin Xu, Qi Zou

**Affiliations:** 1School of Journalism and Information Communication, Huazhong University of Science and Technology, Wuhan 430074, China; d202181505@hust.edu.cn; 2Non-Traditional Security Institute, Huazhong University of Science and Technology, Wuhan 430074, China; lshuai@hust.edu.cn (S.L.); xiaolin@hust.edu.cn (X.X.); 3College of Public Administration, Huazhong University of Science and Technology, Wuhan 430074, China; 4College of Public Administration and Law, Hunan Agricultural University, Changsha 410128, China

**Keywords:** COVID-19, digital transformation, recovery, cities, empirical analysis

## Abstract

Background: Digital transformation has become a key intervention strategy for the global response to the COVID-19 epidemic, and digital technology is helping cities recover from the COVID-19 epidemic. However, the effects of urban digital transformation on the recovery from the COVID-19 epidemic still lack mechanism analyses and empirical testing. This study aimed to explain the theoretical mechanism of urban digital transformation on the recovery from the COVID-19 epidemic and to test its effectiveness using an empirical analysis. Methods: This study, using a theoretical and literature-based analysis, summarizes the impact mechanisms of urban digital transformation on the recovery of cities from the COVID-19 epidemic. A total of 83 large- and medium-sized cities from China are included in the empirical research sample, covering most major cities in China. The ordinary least squares (OLS) method is adopted to estimate the effect of China’s urban digitalization level on population attraction in the second quarter of 2020. Results: The theoretical analysis found that urban digital transformation improves the ability of cities to recover from the COVID-19 epidemic by promoting social communication, collaborative governance, and resilience. The main findings of the empirical analysis show that the digital level of a city has a significant positive effect on urban population attraction (*p* < 0.001). Conclusions: A positive relationship was found between urban digital transformation and the rapid recovery of cities from the COVID-19 epidemic. Digital inventions for social communication, collaborative governance, and urban resilience are an effective way of fighting the COVID-19 emergency.

## 1. Introduction

At the end of 2019, the COVID-19 epidemic broke out globally. As of 24 October 2021, the number of cases of COVID-19 infection worldwide reached 243 million, with 4.9 million deaths [1]. The widespread distribution of COVID-19 has not only hindered the development of the economy but has also brought the production and lives of people across the world to a standstill. Countries worldwide have adopted various epidemic prevention and control measures, such as closing borders, prohibiting gatherings, and isolating infected people, to curb the spread of COVID-19 [2]. This type of large-scale, nonpharmaceutical quarantine intervention reduces population movement and contact, effectively blocks the spread of the virus, and is currently one of the most important measures widely adopted by many countries [3]. However, it cannot be implemented for a long time due to the sacrifice of normal economic and social order [4]. Therefore, the main optimization direction for combating the COVID-19 epidemic worldwide is the use of digital technology to achieve agile governance and adaptive governance in a state of emergency. The digital transformation strategies should be favorable for all aspects of COVID-19 epidemic prevention and control, such as contact tracking, quarantine, health care, education, and the realization of the rapid recovery from the COVID-19 epidemic [5,6].

In this context, many scholars have studied the application of digital technology in the antiepidemic process and the relationship between digitalization and antiepidemic strategies. In terms of digital technology applications, researchers have pointed out that mobile phones are the most widely used digital tool in the process of blocking the spread of COVID-19. Mobile phones can not only be used to collect user movement trajectories, but also become an important carrier for remote medical treatment, remote education, and the disclosure of epidemic information [7]. A typical application is the Geographic Information System (GIS), which can collect the locations and numbers of new cases, deaths, and cured cases, as well as generate real-time epidemic maps to find the source of an outbreak [8]. Moreover, thermal imaging systems, smart sensors, and other Internet of Things (IoT) devices installed in public places and areas with heavy traffic, such as airports, railway stations, and subway stations, can detect suspected cases in real time [9]. The underlying digital architecture formed by the urban digital transformation, including but not limited to digital infrastructure, digital technocrats, digital systems, and digital technology, becomes a vital element in the fight against the epidemic.

Furthermore, urban digital transformation is the main way for the integrated application of digital technology in the field of social governance. Many scholars have discussed the relationship between digital transformation and combating the COVID-19 pandemic, providing indirect and broad-based evidence that digital transformation facilitates urban recovery from the COVID-19 epidemic. Specifically, most researchers believe that urban digital transformation can increase the speed of information dissemination and improve the timeliness of treatment in the early stage of the fight against the epidemic [10]. For example, digital communication strategies can improve the agility of governments’ governance, digital education programs can protect healthy people from infection, and digital patient management can curb the spread of the virus and ensure that confirmed patients receive treatment [11]. In the antiepidemic process, the establishment of a digital platform to integrate epidemic data promotes the sharing of epidemic data among medical, transportation, and community organizations, thereby effectively improving the efficiency of community prevention and control [8]. Epidemic data sharing also provides data support for the generation of real-time epidemic maps, personal health codes, and travel codes, enabling precise prevention and control of the epidemic [12]. When the epidemic is controlled, continuous intelligent monitoring of epidemic data, such as the flow of people and logistics, can normalize the antiepidemic activities and ensure the gradual recovery of society from the impact of the epidemic [9]. In some empirical studies on the impact of urban digital transformation on the antiepidemic effect, Yang and Chong studied the relationship between smart city investment and the number of infected cases in the city [13]. They found that with more investment in smart-city construction, confirmed cases were fewer [13]. Radeef and Rekha proved the negative correlation between the flow of epidemic information on social media in the city and the number of infected cases in the area [14].

The direct effect of urban digital transformation on the recovery of cities from disasters represented by the COVID-19 epidemic has attracted the attention of individual scholars. In terms of the epidemic, the case analysis of Shen et al. showed that digital platforms build a more resilient public service system for cities by promoting public entrepreneurship, inter-organizational coordination, and citizen co-production of public services in the state of COVID-19 emergency [15]. A literature review by Sharifi et al. manifested that urban digital transformation can help build capacity to predict COVID-19 epidemic patterns, facilitate an integrated and timely response, support overstretched sectors, minimize supply chain disruptions, safeguard the continuity of essential services, and provide solutions for optimizing the normal operation of cities in the state of epidemic [16]. In terms of other disasters, a literature review by Sarker et al. revealed that urban digital transformation can improve the speed and effectiveness of linkages between disaster information and system responses, helping to mitigate the risks and impacts of urban socio-ecological vulnerabilities and improve the recovery of cities from disasters [17]. A case study by Hong et al. focused on the urban digital transformation to enable rapid recovery from hurricane disasters, that is, large-scale mobility data can be used to proactively monitor community activities before and during disasters, allowing for near real-time assessment of the impact of disasters, with significant advantages in disaster management and planning [18]. Allaire analyzed the case and found that social media platforms can provide real-time data on flood location and depth, helping cities issue alerts and take action in a timely manner to accelerate recovery from disasters [19].

In general, the impact of urban digital transformation on the rapid recovery of cities from disasters represented by the COVID-19 epidemic is a key issue that is currently receiving attention. However, most studies were mainly focused on concept construction and the status quo of technology application, and the research methods were mostly qualitative research methods such as literature review case analysis. Most issues were concerned with the scenario-based application of digital tools, and few researchers fully focus on the impact of urban digital transformation on the recovery of cities from the COVID-19 epidemic. In the existing rare empirical studies, the focus is on the link between individual urban digital interventions and the number of confirmed cases, and the selected variables cannot comprehensively and accurately measure the level of urban digitization and the recovery of cities from COVID-19. For example, Yang and Chong did not verify the relationship between the amount of investment in smart cities (dependent variable) and the level of smart city construction [13]. Radeef and Rekha could not fully and directly prove the relationship between urban digital transformation and the recovery from the COVID-19 epidemic [14]. Research needs to be carried out from a deeper and more comprehensive perspective to make up for the insufficiency of existing research.

Based on this, we put forward the research question of this paper: What is the theoretical mechanism by which digital transformation affects the recovery of cities from the COVID-19 epidemic, and how can it be quantitatively verified? To answer this question, this study aimed to comprehensively expound the theoretical mechanisms by which urban digital transformation promotes the recovery from the COVID-19 epidemic and to examine the real associations between them using empirical analysis methods. To achieve this research purpose, in the following content, Section 2 explains the theoretical mechanism and proposed research hypotheses. Section 3 presents the materials, variable measures, and main methods of this study. In Section 4, OLS was adopted to verify the positive effect of digital transformation on the rapid recovery of cities from the COVID-19 epidemic. Section 5 discusses the main findings of this study and provided policy and practices recommendations. Section 6 summarizes and refines the full text. Section 7 addresses the limitations of this study.

## 2. Theoretical Basis and Study Hypothesis

A series of positive antiepidemic activities have been adopted worldwide to curb the spread of the epidemic and enable cities with outbreaks to recover in the shortest possible time. Recovery refers to the process of restoring a region to a normal state after a disaster [20]. This normal state does not indicate restoration to the state prior to disaster but the restoration to an acceptable state [21] or to the normal performance level [22]. Previous studies have proven that urban governance capabilities and the level of smart city construction affect urban resilience for epidemic prevention and control [23], and resilience affects the recovery speed of cities after a disaster [23]. Therefore, based on a previous research and theory analysis, the impact path and mechanism of urban digital transformation on the recovery of the COVID-19 epidemic in cities are explained through promoting communication, collaborative governance, and urban resilience.

First, urban digital transformation improves the ability of cities to recover from the epidemic through multiparticipant communication from three aspects, namely, information collection, information transmission, and information sharing. In terms of information collection, each epidemic data service platform is adopted to collect real-time information, such as the number of confirmed cases, the number of cured cases, the trajectory of infected persons, and the list of medium- and high-risk cities. In terms of information transmission, based on the data collected by the epidemic data service platform, traditional and new media workers push the epidemic information to each user through social platforms, such as Twitter, Tik Tok, or online news applications, which not only expand the scope of epidemic information dissemination but also greatly improve its efficiency [24]. At the same time, social media and mobile media provide citizens and authorities with a virtual environment and platform for the communication of health issues, allowing the dissemination of epidemic information and social communication to be further expanded [25,26]. In terms of information sharing, researchers, governments, and companies share important public health, COVID-19 and clinical data through scientific research literature sharing platforms, COVID-19 resource databases, and citizen medical and health platforms. Timely, comprehensive, and transparent information is provided to governments, society, and the public worldwide, and a foundation is laid for multiple social actors to efficiently respond to the epidemic [24].

Second, urban digitalization can help the government accurately provide public services and satisfy citizens’ requirements; improve the performance of urban governance; and promote the coordinated governance of governments, society, and the public in epidemic prevention and control. Specifically, in terms of government-to-government collaboration, urban digital platforms help government agencies overcome the “information gap,” allowing data-driven collaborative decision making to be more extensive in COVID-19 epidemic prevention and control. For example, many local governments have set up portals to share crisis management experience and a major global COVID-19 recovery working group has been established to evaluate different antiepidemic methods [27]. In terms of government-to-business collaboration, digital technology companies have innovated emergency management platforms based on artificial intelligence technology to help governments achieve comprehensive digital contact tracing. In addition, thermal imaging technology and face recognition technology are deployed in public places to rapidly identify high-risk individuals from crowds in China and Singapore [28]. Furthermore, the e-commerce platform and IoT platform support the contactless delivery of medical and living supplies in the state of emergency, greatly ensuring the smooth implementation of mandatory measures, such as isolation and movement restrictions. In terms of government-to-public collaboration, the government relies on online government service platforms to provide public services to selectively and maximally satisfy the requirements of citizens in a state of public health emergency. At the same time, new media, such as Weibo, WeChat, and Toutiao, can be adopted to guide the public to scientifically understand the epidemic, eliminate citizens’ panic, and adhere to accurate measures to fight against the epidemic. Digital interventions also help authorities and the public supervise charitable organizations to disclose social donation information on the network to ensure sufficient emergency supplies.

Urban resilience is a key element in helping the city recover rapidly when it encounters an emergency [29]. Urban resilience refers to the ability of urban systems, social ecology, and social technology networks to adapt to changes in the face of disturbance. The core of urban resilience is that, when unpredictable disasters occur, cities must respond rapidly and effectively to reduce the hazards of disasters [30]. It contains five dimensions, namely, nature, economy, society, materials, and systems. Nature refers to the ecological environment; economy refers to socio-economic development; society refers to the resilience of the community and the general population; materials refer to the resilience of infrastructure; systems refer to the government’s policies in the face of disasters [30]. The impact mechanism of urban digital transformation on urban resilience is mainly introduced from these five aspects.

First, urban digital transformation improves the natural resilience of cities during the COVID-19 epidemic. A set of intelligent systems including sensors, actuators and smart phones, and other IoT devices, which can collect weather, water, and geological data in real time, monitor the actual status of the urban ecosystem and provide an alarm system [31]. Second, urban digital transformation can improve urban economic resilience during the epidemic. Urban digital transformation can promote the construction of a digital infrastructure and the development of digital payments. Thus, digital finance becomes a key measure for many countries in overcoming the economic shocks caused by the epidemic [32]. Third, the development of social platforms and social media has facilitated communication between urban citizens [33], provided more comprehensive public services, and allowed the online and offline integration of people’s production methods and lifestyles. This condition not only improved the quality of life of citizens and their ability to respond to risks during the epidemic but also improved social resilience [34]. Fourth, urban infrastructure construction in the digital age is no longer limited to traditional infrastructure construction. The construction of digital infrastructure, such as broadband, network information systems, cables, and wireless technologies, also becomes a basic task in a state of emergency. The construction of digital infrastructure further increases the resilience of the urban infrastructure to improve the city’s material resilience [35]. Fifth, the COVID-19 epidemic places higher demands on the agility and adaptability of the government. The governments of Singapore, China, and South Korea have selected a path that combines technological governance and coordinated governance to develop government policies that are more timely, targeted, and flexible to curb the spread of COVID-19, thereby improving the systems’ resilience [36,37].

The theoretical background indicates that urban digital transformation can affect the collection, transmission, and sharing of information; promote social organizations to coordinate the management of the epidemic; improve urban resilience, responsiveness, and agility; and help cities recover from the COVID-19 epidemic as rapidly as possible. Therefore, we propose the following hypothesis.

**Hypothesis** **1.**
*Urban digital transformation positively affects the recovery of cities from the COVID-19 epidemic.*


## 3. Materials and Methods

### 3.1. Study Design and Sample Selection

The relationship between the level of digital transformation in major cities in China and the degree of recovery from the COVID-19 epidemic was explored at a specific time. The data are cross-sectional data released by authoritative organizations. Based on the representativeness of cities and the availability of observations of related variables, we selected China’s large and medium-sized cities and cities with a population of more than 3 million released by the National Statistics Bureau of China at the end of 2019 as initial research samples. Then, any alternative cities that lacked the observed values of any variables were excluded in this study [38]. At last, 83 cities in China were included in the research sample. The sample cities are major cities in China, covering the majority of provinces and 45.71% of the population (0.64 billion/1.40 billion) of China at the end of 2019, as shown in Figure 1.

OLS was used to evaluate the impact of the urban digitalization level on the recovery degree of the COVID-19 epidemic in cities in an appropriate period. This condition strengthens the persuasiveness of the need to fight the epidemic through digital transformation.

### 3.2. Independent Variable

The independent variable is the urban digitalization level. Urban digitization means that cities use ICT technology to build various information infrastructures and application capabilities, coupled with urban scene resources, and realize the integration and symbiosis of the “digital world” and the “physical city” [39]. Urban digitalization is manifested in various aspects such as industrial development, urban construction, government management, social governance, and public services.

The urban digitalization level data were obtained from the “2020 China Top 100 Digital Cities Research White Paper” provided by CCID Consulting Co., Ltd, Beijing, China. An analytic hierarchy process was adopted to determine the index system of the urban digital digitization level, which mainly includes six aspects, namely, digital economy, digital governance, digital government, digital people’s livelihood, digital innovation, and digital infrastructure, containing 50 secondary and tertiary indicators [39]. The urban digitalization level of China is a continuous variable, and the evaluation period was 2019. The observed value was not only derived from the “2020 China Top 100 Digital Cities Research White Paper”, but also directly provided by CCID Consulting Co., Ltd, Beijing, China

### 3.3. Dependent Variable

The evaluation indicators of recovery degree in a specific emergency are not uniform across different fields. For example, the indicators for evaluating the level of earthquake recovery mainly include temporary resettlement, permanent housing, schooling, and livelihoods [20]. However, the indicators for evaluating the degree of disaster recovery in tourist cities are mainly economic resilience and tourism resilience [40]. We considered introducing urban population attraction in the second quarter of 2020 as an indicator to appropriately measure the recovery of cities from the COVID-19 epidemic. Urban population attraction is the core manifestation of urban vitality. It is calculated by the ratio of the new inflow of a normal population to a specific city to the average new inflow of a normal population to all cities in China. China’s urban population attraction in the second quarter of 2020 is a continuous variable. The observed value of this variable was obtained from the “Research Report on China’s Urban Vitality in the Second Quarter of 2020” released by Baidu [41]. The second quarter of 2020 was selected as the study period because Wuhan officially lifted the city blockade on 8 April 2020, which marked the initial stage of China’s comprehensive resumption of work and production. Cities with better antiepidemic outcomes during this period can attract more labor. In other words, urban population attraction in this quarter can largely represent the recovery from the urban epidemic [41].

In addition, although incorporating the lagged terms of the dependent variable into the model has the potential to remove the effects of more confounding factors in the general case, we did not consider the lagged terms of the dependent variable mainly due to China’s lockdown measures in Wuhan City on 23 January 2020 and travel restrictions on a large scale. This period was the eve of Chinese New Year, the Spring Festival. The Spring Festival is the most important festival in China, and most of the inter-regional workers return to their hometowns to celebrate before the festival, but they traveled very little with the outbreak of the COVID-19 epidemic. Therefore, once regional lockdowns and nationwide travel restrictions were lifted, cities that had recovered more from the epidemic were initially more likely to attract more workers. We believe that in this special period, the urban population attraction in the second quarter of 2020 can be well used as an indicator of the degree of recovery from the COVID-19 epidemic in China, and its lag effect does not need to be considered.

### 3.4. Control Variable

Urban socio-economical characteristics were incorporated as control variables into the analysis model to avoid the influence of urban social and economic confounding factors on the model. We used the urban hierarchy as the control variable, which can reflect the comprehensive urban social and economic development level after processing. This is mainly divided into first-tier cities, second-tier cities, third-tier cities, fourth-tier cities, and fifth-tier cities. They are, respectively, valued as: first-tier cities = 1, second-tier cities = 2, third-tier cities = 3, fourth-tier cities = 4, and fifth-tier cities = 5. This means that the lower the urban hierarchy value, the higher the urban comprehensive economic and social development level. The classification of city levels was obtained from the “2020 City Business Charm Ranking List” released by China Business News [42]. The leaderboard generation method is a hierarchical analysis method. Experts from the Institute of New First-tier Cities were invited to assign weights from five aspects, namely, business, transportation, population, life, and development potential to arrive at the final ranking result [42].

### 3.5. Statistical Analyses

First, we used MS Excel 2016 and SPSS 23.0 to enter and clean the relevant data. Subsequently, the strata SE 15 software package was adopted for descriptive statistical analysis, Pearson’s correlation analysis, and OLS. Descriptive statistics were used to analyze independent variables (urban digitalization level), dependent variables (urban population attraction), and control variables (urban hierarchy). Pearson’s correlation analysis was mainly used to analyze the correlation among the three variables. OLS was used to verify the positive effect of the urban digitalization level on the urban population attraction. The test efficiency of this study was α = 0.05.

## 4. Results

### 4.1. Descriptive Statistics of Urban Digitalization, Urban Population Attraction, and Urban Hierarchy

Table 1 shows the characteristics of the urban population attraction, the urban digitalization level, and the urban hierarchy of the sample cities. Specifically, the urban population attraction ranges from 0.43 to 16.89, with an average of 3.46 and an SD of 3.58, indicating that the urban population attraction has a large gap between the sample cities, and the overall level is low. The urban digitalization level ranges from 55.02 to 89.36. The average is 69.26, and the SD value is 8.04. This finding shows that although there are gaps in the level of digitization between cities in China, these gaps are smaller than those in the level of population attraction. Additionally, the overall digitalization level of Chinese cities is at a relatively mature stage. This also explains why China has had the ability to deploy digital interventions on a large scale during the COVID-19 outbreak. The urban hierarchy ranges from 1 to 4. That is, all sample cities are Tier 1 to Tier 4 cities. The average is 2.22, and the SD is 0.84, indicating that the sample cities are evenly distributed among different hierarchies. Tier 5 cities were not included in the sample because the cities selected for this study are major cities—large- and medium-sized cities from China. The Tier 5 cities are generally small-sized cities, so they were not included in the sample.

### 4.2. Correlation Analysis of Urban Digitalization, Urban Population Attraction, and Urban Hierarchy

To preliminarily test whether the changing trends among the urban digitalization level, urban population attraction, and urban hierarchy are consistent, Pearson’s correlation analysis was used to test their correlation. The results are shown in Table 2. Specifically, the urban digitalization level was significantly positively correlated with urban population attraction (*r* = 0.79, *p* < 0.001). The urban digitalization level has a negative correlation with urban hierarchy (r = −0.83, *p* < 0.001). Urban population attraction has a negative correlation with urban hierarchy (*r* = −0.74, *p* < 0.001). In other words, the urban digitalization level and urban population attraction have the same changing trend. Meanwhile, a lower urban economic and social development hierarchy indicates a lower urban digitalization level and urban population attraction.

### 4.3. Validation Analysis of the Impact of Urban Digitalization on Urban Population Attraction

We used OLS for analysis to verify the positive effect of the urban digitalization level on urban population attraction. The variance inflation factor (*vif*) values of the independent variable and control variables were within the traditional acceptable range of less than 10, and most of them satisfied the more stringent acceptable threshold of 5. Therefore, the OLS model used in this study can be considered free of multicollinearity.

The results of OLS are shown in Table 3. The table shows that the urban digitalization level has a significant positive effect on urban population attraction (*Coef.* = 0.21, *p* < 0.001, 95%CI = [0.11, 0.32]). From the perspective of model explanatory power, the *R*^2^ of this model is 0.69, showing that the independent variable and the control variable have high explanatory power for the dependent variable, and the model is relatively robust. Therefore, the higher urban digitalization level indicates higher urban population attraction, indicating better recovery of the city. This finding verified the hypothesis of this study.

## 5. Discussion

During the COVID-19 epidemic, many countries and regions have attempted to improve their antiepidemic capabilities through urban digital transformation. The importance of digital transformation in the epidemic is self-evident, but the vast majority of the research is in the form of qualitative analyses summarizing the experiences. Empirical studies on the impact of digital transformation on the effectiveness of epidemic prevention and control are limited. Therefore, this research aimed to analyze the impact of urban digital transformation on the rapid recovery of cities from the COVID-19 epidemic and use data from 83 major cities in China to conduct an empirical analysis. The main findings of this study are as follows.

First, by combing through previous studies, we found that urban digital transformation can promote social communication, coordinated governance, and urban resilience, thereby improving the effectiveness of epidemic prevention and control. Subsequently, the hypothesis that digital transformation promotes the recovery of cities from the COVID-19 epidemic is proposed. Previous qualitative research has focused on the mechanisms by which the specific fields of urban digital transformation, such as a limited number of digital public health interventions, facilitate the specific fields of urban recovery such as the continuity of essential services from the COVID-19 epidemic [15,16]. In contrast, this study provided deeper and more comprehensive insights into a broader urban digital transformation that aids comprehensive recovery from the COVID-19 epidemic.

Second, we used urban population attraction in the second quarter of 2020 to assess the degree of urban recovery degree from the epidemic and OLS to verify that the urban digitalization level positively affects urban population attraction. The study hypothesis, that is, a higher urban digitalization level indicates better recovery from the epidemic, was quantitatively verified. Previous empirical studies similar to this study focused on the association between a certain aspect of urban digital transformation such as smart city investment/the flow of epidemic information on social media and the number of confirmed cases [13,14] and were insufficient to fully and directly verify the association between urban digital transformation and recovery from the COVID-19 epidemic. Our empirical study filled the gap in this field by more comprehensively and scientifically measuring the urban digitalization level and the recovery of cities from the epidemic.

In reality, the digital transformation of Chinese cities plays an important role in the prevention and control of the COVID-19 epidemic. This finding is due to the fact that the Chinese government has always adhered to the digital transformation strategy and has continued to build a digital government throughout the 21st century. Prior to the COVID-19 outbreak, China’s central and local governments had established dedicated government data management departments and big-data platforms [5]. The purpose was to integrate and manage the data of all the government departments and form a cross-level and cross-regional comprehensive government data management mechanism. These efforts continue to penetrate the field of public health, providing an organizational and technical foundation for digital antiepidemic strategies. For example, in 2004, China established the largest direct network reporting system for infectious diseases and public health emergencies. It was comprehensively promoted in the national medical and health systems. This system was subsequently upgraded with an automatic early warning subsystem and a GIS system, which enabled nationwide epidemic monitoring, early warning, tracking, and coordinated management [43,44]. It shortened the time taken to detect an epidemic from five days to four hours. At the same time, the Chinese central government issued a number of big-data application standards in the medical and health fields. This condition ensures that the antiepidemic digital infrastructure is sufficient, promotes the work process of data sharing and information disclosure, and lays the foundation for the preparation of epidemic dynamic monitoring and comprehensive risk assessment [45,46]

Although digital tools have been widely used during the COVID-19 outbreak, security issues, such as personal privacy, have required urgent resolutions [47]. The risk issues, legal boundaries, and moral bottom line of digital technologies, such as symptom monitoring, isolation, and traffic modeling, remain to be discussed [48]. Cybercrime and the proliferation of digital false information have affected public trust in digitization, resulting in digital destruction [49]. Therefore, in the process of urban digital transformation, personal data security should be firmly maintained, and data processing standards should be observed [50]. At the same time, digital transformation cannot be transformed into “digitalism”. The government should use forward-looking policy innovations to guide digital transformation with respect to health, science, and democracy [6]. Moreover, decentralized data collection methods should be used to deal with data proliferation, data deprivation, and digital divide issues [51], while improving digital platform management methods. This condition can reshape the trust of citizens and allow digital tools to play a greater role in the prevention and control of the epidemic [47,52].

Finally, this study provides policy and management implications for local governments to take advantage of the opportunity of urban digital transformation to achieve rapid recovery from the epidemic. That is, cities need to develop a digital strategy for the epidemic, embedding the urban digital transformation into public health interventions and emergency management systems. Specifically, first, cities should technically establish a digital platform that includes functions such as quarantine measures, surveillance and early warning, contacts tracing, treatment, communication, decision-making, and recovery. Second, and more importantly, cities should take advantage of the opportunities of digital transformation to achieve a more holistic, inclusive, multi-participatory, and public-centered crisis governance model at the organizational level. Finally, cities need to leverage digital transformation to implement functional designs including improved social communication, collaborative governance, and urban resilience for the epidemic emergency.

## 6. Conclusions

With the COVID-19 epidemic still spreading globally, more precise, agile, and resilient interventions need to be designed for the rapid recovery of cities, and digital transformation is expected to realize this vision. However, relevant qualitative studies have focused on concept construction and the status quo of technology application and only discussed the extremely limited digital intervention scenarios [15,16]. Research whose indicators are selected by a small number of empirical studies cannot fully reflect the direct relationship between digital transformation and the recovery of cities from the COVID-19 epidemic [13,14]. How to comprehensively expound on how cities use digital transformation to achieve the rapid recovery from the COVID-19 epidemic from a theoretical level, and what kind of objective and reasonable data can be adopted to prove whether there is a positive relationship between the two, become pressing challenges that must be addressed to design agile, fully functional, low-cost, and scalable digital interventions for countries and areas that are being hit by the epidemic.

To address these issues and overcome the limitations of existing fragmented and inadequate studies, this study provided novel insights into digital interventions for global recovery from the COVID-19 epidemic, especially by innovatively using different fields of theory and more robust empirical analysis to strengthen the evidence for digital interventions in the epidemic. Specifically, this study elucidated the theoretical mechanism of how digital transformation facilitates the recovery from the COVID-19 epidemic from a deeper and more comprehensive perspective, and for the first time directly validates their positive association. The results of this study showed that the urban digital transformation improves the recovery of cities from the COVID-19 epidemic by promoting social communication, collaborative governance, and urban resilience. The empirical analysis of major cities in China verifies that urban digital transformation positively affects the improvement of the cities’ ability to recover from the epidemic. This research had a dual contribution at both theoretical and practical levels. At the theoretical level, this study integrated classic governance theories such as social communication, collaborative governance, and resilience governance into digital transformation to promote the recovery of cities from the COVID-19 epidemic. This is conducive to building a resilient social economic governance, and natural environment in the digital age to help cities effectively respond to an epidemic that arises anytime and anywhere, and provides a complete set of top-level design principles for epidemic digital interventions. At a practical level, to our knowledge, this study is the first to use empirical research to demonstrate that digital transformation has a direct catalytic effect on the recovery of cities from the COVID-19 epidemic. It provides strong reasons for cities around the world to take advantage of the opportunities presented by digital transformation to more aggressively and extensively develop digital interventions to combat the COVID-19 epidemic.

At last, this study provided several policies and management implications in digital interventions for global recovery from the COVID-19 epidemic. That is, cities need to rely on digital transformation to build a complete digital platform for epidemic emergency management, and integrate digital technology systems with governance systems more perfectly, especially leveraging digital transformation to achieve key digital intervention functions such as better information sharing and social communication, more integrated and collaborative governance, and more resilient cities. This provided new and crucial management implications for countries around the world to build smarter, more agile, and more integrated epidemic emergency management systems. As a condition for the success of these digital interventions, government policy should also focus on balancing digital ethics and public health, especially to protect public privacy and eliminate the digital divide.

## 7. Limitations and Future Research

Some limitations exist in this study. First, due to the limitations of the data sources, the measurement period of the urban digitization level in this study was in years, while the measurement period of urban population attractiveness was in quarters, which may blur the relationship between the two to a certain extent. Second, given the limited number of variables and the sample size, the model still contains bias, which reduces the accuracy of the results. Further, future research needs to explore in more detail how various digital interventions are combined with the urban epidemic emergency management system, and use empirical analysis to further verify their actual impact on the recovery from the epidemic, so as to provide more targeted evidence for establishing a more effective, smart, agile, intelligent, and integrated emergency management system.

## Figures and Tables

**Figure 1 ijerph-19-03567-f001:**
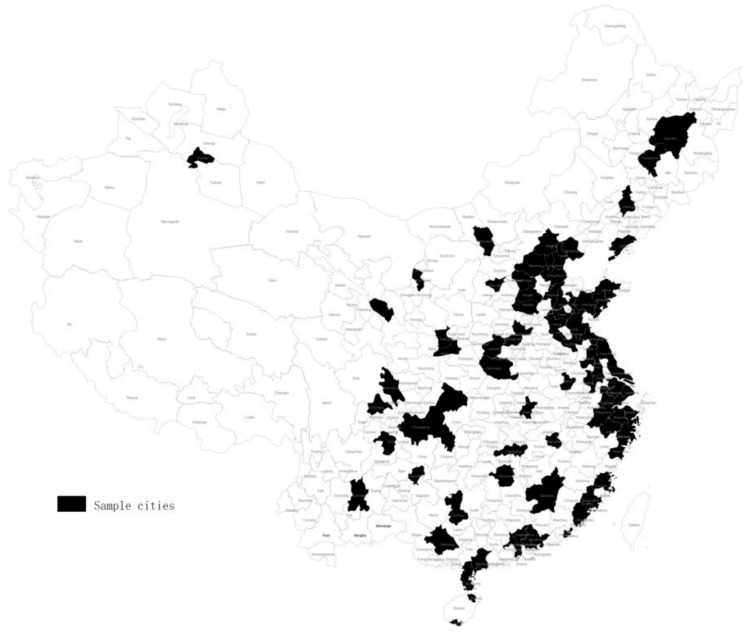
Sample cities in this study.

**Table 1 ijerph-19-03567-t001:** Descriptive statistics of the variables.

Variables	*n*	Mean (SD)	Min	Max
Urban population attraction	83	3.46 (3.58)	0.43	16.89
Urban digitalization level	83	69.26 (8.04)	55.02	89.36
Urban hierarchy	83	2.22 (0.84)	1	4

**Table 2 ijerph-19-03567-t002:** Correlation of variables.

Variables	Urban Population Attraction	Urban Digitalization Level	Urban Hierarchy
Urban population attraction	1		
Urban digitalization level	0.79 ***	1	
Urban hierarchy	−0.74 ***	−0.83 ***	1

Note: *** means *p* < 0.001.

**Table 3 ijerph-19-03567-t003:** Analysis of the effect of the urban digitalization level on urban population attraction.

Variables	*Coef.*	*Std. Err.*	95% Confidence Interval	*Std. Coef.*	*t*	*P*	*vif*
Independent variable							
Urban digitalization level	0.21	0.05	[0.11, 0.32]	0.48	4.18	<0.001	3.33
Control variables							
Urban hierarchy (base 1)							
2	−3.03	0.78	[−4.58, −1.47]	−0.41	−3.88	<0.001	2.77
3	−3.6	1.05	[−5.70, −1.50]	−0.49	−3.41	<0.001	5.14
4	−3.32	1.66	[−6.62, −0.01]	−0.17	−2	0.049	1.90
Constant	−8.86	4.13	[−17.08, −0.64]		−2.15	0.035	

## Data Availability

The datasets used and analyzed in the current study are available from the corresponding author on reasonable request.

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
