# Peer review of "Can Digital Transformation Promote the Rapid Recovery of Cities from the COVID-19 Epidemic? An Empirical Analysis from Chinese Cities"

_ijerph, 2022, doi:10.3390/ijerph19063567_

Round 1
Reviewer 1 Report
The topic is, in principle, interesting and within the scope of International Journal of Environmental Research and Public Health. It explains the theoretical mechanism of urban digital transformation in combating the COVID-19 epidemic and tests its effectiveness using an empirical analysis with data from 83 major cities in China. The main added value of the article is that it summarizes the impact mechanisms of urban digital transformation on the recovery of cities from the COVID-19 epidemic while most of the current research is in the form of qualitative analyses summarizing experiences.
There are some problems of the paper in its present state, especially in the Discussion and Conclusion Sections. The authors need to introduce modifications and clarifications in the manuscript and answer to a few comments.
- Article's structure.
Please explain in Section 1 “Introduction” how the paper is structured / organized.
- Written Expression-English
There are some problems with written expressions throughout the manuscript. The paper should be looked at and improved from the language perspective. A example: Lines 328 and 329, the word “indicates” is used 3 times.
- Definition/Description of paper objectives
Please develop a specific research question that responds to the hypothesis raised in the study to clarify the research.
- Literature Review
Section 2 is generally well-written and organized. This part demonstrates an adequate understanding of the impact of urban digitalization.
- Discussion
There is a lack of literature to support the results obtained and the authors fail to further discuss the results.
Authors should include more relevant papers in this section to support the results and to answers the hypothesis stated in the manuscript.
- Conclusions and managerial implications
I strongly suggest to improve the Conclusions Section and to add possible policy implications of the results obtained. How these results can be used to promote digital transformation for the recovery of COVID-19 pandemic around the world.
Conclusions should state the value added of your paper, the contribution compared to available literature. What are the managerial implications in these cities of your results?
I also strongly suggest inserting possible future lines of research in this section.
- Minor Errors
There are some minor errors that should be corrected, see lines 291 (insert a dot at the end of the sentence), 388 (there are two dots in the sentence)
Author Response
Dear reviewers:
Thank you very much for your letter and for the comments on our manuscript entitled “Can digital transformation promote the rapid recovery of cities from the COVID-19 epidemic? An empirical analysis from Chinese cities” (Submission ID: ijerph-1566936). Those comments are not only valuable and helpful for revising and improving this paper, but also a priceless reference to our future researches. We have studied comments carefully and accordingly made corrections that we hope could meet with approval standards. To help the readers better understand this study, the language of the manuscript has been sent to be edited by the MDPI Author Services (https://www.mdpi.com/authors/english). The main corrections in the paper and the responds to the reviewers’ comments are as following:
Total comments: The topic is, in principle, interesting and within the scope of International Journal of Environmental Research and Public Health. It explains the theoretical mechanism of urban digital transformation in combating the COVID-19 epidemic and tests its effectiveness using an empirical analysis with data from 83 major cities in China. The main added value of the article is that it summarizes the impact mechanisms of urban digital transformation on the recovery of cities from the COVID-19 epidemic while most of the current research is in the form of qualitative analyses summarizing experiences.
Point 1: Article's structure. Please explain in Section 1 “Introduction” how the paper is structured/organized.
Response 1: Thank you very much for these valuable comments, and we have added the article’s structure to the Introduction section. The additions are as follows.
“To achieve this research purpose, in the following content, Chapter 2 explained the theoretical mechanism and proposed research hypotheses. Chapter 3 presented the materials, variable measures and main methods of this study. In Chapter 4, OLS was adopted to verify the positive effect of digital transformation on the rapid recovery of cities from the COVID-19 epidemic. Chapter 5 discussed the main findings of this study and provided policy and practices recommendations. Chapter 6 summarized and refines the full text. Chapter 7 addressed the limitations of this study and proposed future research directions.” (see Line 145-151 for details)
Point 2: Definition/Description of paper objectives
Please develop a specific research question that responds to the hypothesis raised in the study to clarify the research.
Response 2: Thank you very much for these valuable comments, and the research question and objectives have been added to the main text. The additions are as follows.
“Based on this, we put forward the research question of this paper: What is the theoretical mechanism by which digital transformation affects the recovery of cities from the COVID-19 epidemic, and how can it be quantitatively verified? To answer this question, this study aimed to comprehensively expound the theoretical mechanisms by which urban digital transformation promotes the recovery from the COVID-19 epidemic, and to examine the real associations between them using empirical analysis methods.” (see Line 139-144 for details)
Point 3: Literature Review
Section 2 is generally well-written and organized. This part demonstrates an adequate understanding of the impact of urban digitalization.
Response 3: Thank you for your affirmation, and we will continue to improve the quality of the manuscript until the last moment.
Point 4: Discussion
There is a lack of literature to support the results obtained and the authors fail to further discuss the results. Authors should include more relevant papers in this section to support the results and to answers the hypothesis stated in the manuscript.
Response 4: Thank you very much for these valuable comments, and we further strengthen the discussion of the results, especially introducing relevant papers for comparative analysis to enhance the persuasiveness of the discussion section. The specific modifications are as follows.
“First, by combing through previous studies, we found that urban digital transformation can promote social communication, coordinated governance, and urban resilience, thereby improving the effectiveness of epidemic prevention and control. Subsequently, the hypothesis that digital transformation promotes the recovery of cities from the COVID-19 epidemic is proposed. In contrast to previous qualitative research focused on the mechanisms by which the specific fields of urban digital transformation such as a limited number of digital public health interventions facilitates he specific fields of urban recovery such as the continuity of essential services from the COVID-19 epidemic[16,17], this study provided deeper and comprehensive insights into a broader urban digital transformation that aids comprehensive recovery from COVID-19 epidemic.
Second, we used urban population attraction in the second quarter of 2020 to assess the degree of urban recovery degree from the epidemic and OLS to verify that the urban digitalization level positively affects urban population attraction. The study hypothesis, that is, a higher urban digitalization level indicates better recovery from the epidemic, was quantitatively verified. Previous empirical studies similar to this study focused on the association between a certain aspect of urban digital transformation such as smart city investment/the flow of epidemic information on social media and the number of confirmed cases[14,15], and were insufficient to fully and directly verify the association between urban digital transformation and recovery from the COVID-19 epidemic. Our empirical study filled the gap in this field by more comprehensively and scientifically measuring urban digitalization level and the recovery of cities from the epidemic.” (see Line 392-412 for details)
Point 5: Conclusions and managerial implications
I strongly suggest to improve the Conclusions Section and to add possible policy implications of the results obtained. How these results can be used to promote digital transformation for the recovery of COVID-19 pandemic around the world.
Conclusions should state the value added of your paper, the contribution compared to available literature. What are the managerial implications in these cities of your results?
I also strongly suggest inserting possible future lines of research in this section.
Response 5: Thank you very much for these valuable comments, and we recognize that the research findings have important policy and management implications for the recovery of the COVID-19 epidemic. Therefore, we have added relevant content to the article. Considering the format of the journal's recent publications and comments from other reviewers, we have included policy and management implications in the Discussion section and future research in the “Limitations and future research” section. A comparative analysis of relevant literature is shown in Response 4. The additions are as follows.
Policy and management implications:
“Finally, this study provides policy and management implications for local governments to take advantage of the opportunity of urban digital transformation to achieve rapid recovery from the epidemic. That is, cities need to develop a digital strategy for the epidemic, embedding the urban digital transformation into public health interventions and emergency management systems. Specifically, first, cities should technically establish a digital platform that includes functions such as quarantine measures, surveillance and early warning, contacts tracing, treatment, communication, decision-making, and recovery. Second, and more importantly, cities should take advantage of the opportunities of digital transformation to achieve a more holistic, inclusive, multi-participatory and public-centered crisis governance model at the organizational level. Finally, cities need to leverage digital transformation to implement functional designs including improved social communication, collaborative governance, and urban resilience for the epidemic emergency.” (see Line 447-458 for details)
Future research:
“Further, future research needs to explore in more detail how various digital interventions are combined with the urban epidemic emergency management system, and use empirical analysis to further verify their actual impact on the recovery from epidemic, so as to provides more targeted evidence for establishing a more effective, smart, agile, intelligent, and integrated emergency management system.” (see Line 474-478 for details)
Point 6:Minor Errors
There are some minor errors that should be corrected, see lines 291 (insert a dot at the end of the sentence), 388 (there are two dots in the sentence).
Response 6: Thank you for pointing out this problem. And we have revised this section and double-checked to correct similar problems in the text.
Reviewer 2 Report
The topic this paper addresses, in the current situation we are living in, is of great relevance. On the one hand, it provides information on which factors help cities overcome the situation caused by the COVID-19; on the other hand, these results can be of great use for the recovery of other contexts.
Some minor changes are recommended. First, in the Discussion section, it is suggested that the results of the present study are linked with scientific studies that have investigated the topic of city recovery. It would be interesting that the authors state in which way their results support or differ from previous research. Secondly, the paragraph from lines 351-369 does not reference any scientific publications. It is suggested that scientific research is cited in the mentioned lines.
Author Response
Dear reviewers:
Thank you very much for your letter and for the comments on our manuscript entitled “Can digital transformation promote the rapid recovery of cities from the COVID-19 epidemic? An empirical analysis from Chinese cities” (Submission ID: ijerph-1566936). Those comments are not only valuable and helpful for revising and improving this paper, but also a priceless reference to our future researches. We have studied comments carefully and accordingly made corrections that we hope could meet with approval standards. To help the readers better understand this study, the language of the manuscript has been sent to be edited by the MDPI Author Services (https://www.mdpi.com/authors/english). The main corrections in the paper and the responds to the reviewers’ comments are as following:
Total comments: The topic this paper addresses, in the current situation we are living in, is of great relevance. On the one hand, it provides information on which factors help cities overcome the situation caused by the COVID-19; on the other hand, these results can be of great use for the recovery of other contexts.
Point 1:Some minor changes are recommended. First, in the Discussion section, it is suggested that the results of the present study are linked with scientific studies that have investigated the topic of city recovery. It would be interesting that the authors state in which way their results support or differ from previous research.
Response 1: Thank you very much for these valuable comments, and we have added a comparative analysis with previous studies in the Results section and accordingly detailed the relevant literature in the Introduction section. We have added a comparative analysis with previous studies to the Results section. Meanwhile, in order to keep the content coherent, we correspondingly introduce the relevant literature in detail in the Introduction section. The specific modifications are as follows.
Results section:
“First, by combing through previous studies, we found that urban digital transformation can promote social communication, coordinated governance, and urban resilience, thereby improving the effectiveness of epidemic prevention and control. Subsequently, the hypothesis that digital transformation promotes the recovery of cities from the COVID-19 epidemic is proposed. In contrast to previous qualitative research focused on the mechanisms by which the specific fields of urban digital transformation such as a limited number of digital public health interventions facilitates he specific fields of urban recovery such as the continuity of essential services from the COVID-19 epidemic[16,17], this study provided deeper and comprehensive insights into a broader urban digital transformation that aids comprehensive recovery from COVID-19 epidemic.
Second, we used urban population attraction in the second quarter of 2020 to assess the degree of urban recovery degree from the epidemic and OLS to verify that the urban digitalization level positively affects urban population attraction. The study hypothesis, that is, a higher urban digitalization level indicates better recovery from the epidemic, was quantitatively verified. Previous empirical studies similar to this study focused on the association between a certain aspect of urban digital transformation such as smart city investment/the flow of epidemic information on social media and the number of confirmed cases[14,15], and were insufficient to fully and directly verify the association between urban digital transformation and recovery from the COVID-19 epidemic. Our empirical study filled the gap in this field by more comprehensively and scientifically measuring urban digitalization level and the recovery of cities from the epidemic.” (see Line 392-412 for details)
Introduction section:
“The direct effect of urban digital transformation on the recovery of cities from disasters represented by the COVID-19 epidemic has attracted the attention of individual scholars. In terms of the epidemic, the case analysis of Shen et al. showed that digital platforms build a more resilient public service system for cities by promoting public entrepreneurship, interorganizational coordination, and citizen coproduction of public services in the state of COVID-19 emergency[15]. A literature review by Sharifi et al. manifested that urban digital transformation can help build capacity to predict COVID-19 epidemic patterns, facilitate an integrated and timely response, support overstretched sectors, minimize supply chain disruptions, safeguard the continuity of essential services, and provide solutions for optimizing the normal operation of cities in the state of epidemic[17]. In terms of other disasters, a literature review by Sarker et al. revealed that urban digital transformation can improve the speed and effectiveness of linkages between disaster information and system responses, helping to mitigate the risks and impacts of urban socio-ecological vulnerabilities and improve the recovery of cities from disasters[18]. A case study by Hong et al. focused on the urban digital transformation to enable rapid recovery from hurricane disasters, i.e., large-scale mobility data can be used to proactively monitor community activities before and during disasters, allowing for near real-time assessment of the impact of disasters impact, with significant advantages in disaster management and planning [19]. Allaire analyzed the case and found that social media platforms can provide real-time data on flood location and depth, helping cities issue alerts and take action in a timely manner to accelerate recovery from disasters [20].” (see Line 102-121 for details)
Point 2: Secondly, the paragraph from lines 351-369 does not reference any scientific publications. It is suggested that scientific research is cited in the mentioned lines.
Response 2: Thank you very much for these valuable comments, and we cited relevant references in our discussion of the results (see Response 7 for details). In addition, we also cited some references to enhance the scientific nature of the rest of the Discussion section. The specific modifications are as follows.
“In reality, the digital transformation of Chinese cities plays an important role in the prevention and control of the COVID-19 epidemic. This finding is due to the fact that the Chinese government has always adhered to the digital transformation strategy and has continued to build a digital government throughout the 21st century. Prior to the COVID-19 outbreak, China’s central and local government had established dedicated government data management departments and big-data platforms [5]. The purpose was to integrate and manage the data of all the government departments and form a cross-level and cross-regional comprehensive government data management mechanism. These efforts continue to penetrate the field of public health, providing an organizational and technical foundation for digital antiepidemic strategies. For example, in 2004, China established the largest direct network reporting system for infectious diseases and public health emergencies. It was comprehensively promoted in the national medical and health systems. This system was subsequently upgraded with an automatic early warning subsystem and a GIS system, which enabled nationwide epidemic monitoring, early warning, tracking, and coordinated management [44,45]. It shortened the time taken to detect an epidemic from five days to four hours. At the same time, the Chinese central government issued a number of big-data application standards in the medical and health fields. This condition ensures that the antiepidemic digital infrastructure is sufficient, promotes the work process of data sharing and information disclosure, and lays the foundation for the preparation of epidemic dynamic monitoring and comprehensive risk assessment [46,47].” (see Line 413-432 for details)
Reviewer 3 Report
Dear author(s),
Thank you for the interesting article.
Following are the suggestions:
Introduction section shall end with the aim of the manuscript.
The original contribution (scientific and academic) should be stated clear and loud.
Literature should be extended.
Hypothesis should be clearly stated.
Limitations should be written after conclusion section
Managerial implications are missing from the work.
Author Response
Dear reviewers:
Thank you very much for your letter and for the comments on our manuscript entitled “Can digital transformation promote the rapid recovery of cities from the COVID-19 epidemic? An empirical analysis from Chinese cities” (Submission ID: ijerph-1566936). Those comments are not only valuable and helpful for revising and improving this paper, but also a priceless reference to our future researches. We have studied comments carefully and accordingly made corrections that we hope could meet with approval standards. To help the readers better understand this study, the language of the manuscript has been sent to be edited by the MDPI Author Services (https://www.mdpi.com/authors/english). The main corrections in the paper and the responds to the reviewers’ comments are as following:
Point 1: Introduction section shall end with the aim of the manuscript.
Response 1: Thank you very much for these valuable comments, and the aim of the manuscript has been added to the Introduction section. The additions are as follows.
“To answer this question, this study aimed to comprehensively expound the theoretical mechanisms by which urban digital transformation promotes the recovery from the COVID-19 epidemic, and to examine the real associations between them using empirical analysis methods.” (see Line 141-144 for details)
Point 2:The original contribution (scientific and academic) should be stated clear and loud.
Response 2: Thank you very much for these valuable comments, and the aim of the original contribution has been added to the Discussion section. The additions are as follows.
“In contrast to previous qualitative research focused on the mechanisms by which the specific fields of urban digital transformation such as a limited number of digital public health interventions facilitates he specific fields of urban recovery such as the continuity of essential services from the COVID-19 epidemic[16,17], this study provided deeper and comprehensive insights into a broader urban digital transformation that aids comprehensive recovery from COVID-19 epidemic.” (see Line 496-401 for details)
“Previous empirical studies similar to this study focused on the association between a certain aspect of urban digital transformation such as smart city investment/the flow of epidemic information on social media and the number of confirmed cases[14,15], and were insufficient to fully and directly verify the association between urban digital transformation and recovery from the COVID-19 epidemic. Our empirical study filled the gap in this field by more comprehensively and scientifically measuring urban digitalization level and the recovery of cities from the epidemic.” (see Line 406-412 for details)
“Finally, this study provides policy and management implications for local governments to take advantage of the opportunity of urban digital transformation to achieve rapid recovery from the epidemic. That is, cities need to develop a digital strategy for the epidemic, embedding the urban digital transformation into public health interventions and emergency management systems. Specifically, first, cities should technically establish a digital platform that includes functions such as quarantine measures, surveillance and early warning, contacts tracing, treatment, communication, decision-making, and recovery. Second, and more importantly, cities should take advantage of the opportunities of digital transformation to achieve a more holistic, inclusive, multi-participatory and public-centered crisis governance model at the organizational level. Finally, cities need to leverage digital transformation to implement functional designs including improved social communication, collaborative governance, and urban resilience for the epidemic emergency.” (see Line 447-458 for details)
Point 3: Literature should be extended.
Response 3: Thank you very much for these valuable comments, and we have refined, comparatively analyzed and summarized the relevant literature based on your comments to further increase the background and depth of this research. The additions are as follows.
“The direct effect of urban digital transformation on the recovery of cities from disasters represented by the COVID-19 epidemic has attracted the attention of individual scholars. In terms of the epidemic, the case analysis of Shen et al. showed that digital platforms build a more resilient public service system for cities by promoting public entrepreneurship, interorganizational coordination, and citizen coproduction of public services in the state of COVID-19 emergency[15]. A literature review by Sharifi et al. manifested that urban digital transformation can help build capacity to predict COVID-19 epidemic patterns, facilitate an integrated and timely response, support overstretched sectors, minimize supply chain disruptions, safeguard the continuity of essential services, and provide solutions for optimizing the normal operation of cities in the state of epidemic[17]. In terms of other disasters, a literature review by Sarker et al. revealed that urban digital transformation can improve the speed and effectiveness of linkages between disaster information and system responses, helping to mitigate the risks and impacts of urban socio-ecological vulnerabilities and improve the recovery of cities from disasters[18]. A case study by Hong et al. focused on the urban digital transformation to enable rapid recovery from hurricane disasters, i.e., large-scale mobility data can be used to proactively monitor community activities before and during disasters, allowing for near real-time assessment of the impact of disasters, with significant advantages in disaster management and planning [19]. Allaire analyzed the case and found that social media platforms can provide real-time data on flood location and depth, helping cities issue alerts and take action in a timely manner to accelerate recovery from disasters [20].
In general, the impact of urban digital transformation on the rapid recovery of cities from disasters represented by the COVID-19 epidemic is a key issue that is currently receiving attention. However, most studies were mainly focused on concept construction and status quo combing, and the research methods were mostly qualitative research methods such as literature review case analysis. Most issues were concerned with the scenario-based application of digital tools, and few researchers fully focus on the impact of urban digital transformation on the recovery of cities from the COVID-19 epidemic. In the existing rare empirical studies, the focus is on the link between individual urban digital interventions and the number of confirmed cases, and the selected variables cannot comprehensively and accurately measure the level of urban digitization and the recovery of cities from the COVID-19. For example, Yang and Chong did not verify the relationship between the amount of investment in smart cities (dependent variable) and the level of smart city construction [14]. Radeef and Rekha cannot fully and directly prove the relationship between urban digital transformation and the recovery from the COVID-19 epidemic [15]. Research needs to be carried out from a deeper and comprehensive perspective to make up for the insufficiency of existing research.” (see Line 102-138 for details)
Point 4: Hypothesis should be clearly stated.
Response 4: Thank you very much for these valuable comments, and we have re-stated the assumptions clearly. The specific modifications are as follows.
“Hypothesis: urban digital transformation positively affects the recovery of cities from the COVID-19 epidemic.” (see Line 246-247 for details)
Point 5: Limitations should be written after conclusion section
Response 5: Thank you very much for these valuable comments, and we have written Limitations after the Conclusion section. The specific modifications are as follows.
“7. Limitations and future research
Some limitations exist in this study. First, due to the limitations of the data sources, the measurement period of the urban digitization level in this study was in years, while the measurement period of urban population attractiveness was in quarters, which may blur the relationship between the two to a certain extent. Second, given the limited number of variables and the sample size, the model still contains bias, which reduces the accuracy of the results. Further, future research needs to explore in more detail how various digital interventions are combined with the urban epidemic emergency management system, and use empirical analysis to further verify their actual impact on the recovery from epidemic, so as to provides more targeted evidence for establishing a more effective, smart, agile, intelligent, and integrated emergency management system.” (see Line 468-478 for details)
Point 6: Managerial implications are missing from the work.
Response 6: Thank you very much for these valuable comments, and the managerial implications have been added to the Discussion section. The additions are as follows.
“Finally, this study provides policy and management implications for local governments to take advantage of the opportunity of urban digital transformation to achieve rapid recovery from the epidemic. That is, cities need to develop a digital strategy for the epidemic, embedding the urban digital transformation into public health interventions and emergency management systems. Specifically, first, cities should technically establish a digital platform that includes functions such as quarantine measures, surveillance and early warning, contacts tracing, treatment, communication, decision-making, and recovery. Second, and more importantly, cities should take advantage of the opportunities of digital transformation to achieve a more holistic, inclusive, multi-participatory and public-centered crisis governance model at the organizational level. Finally, cities need to leverage digital transformation to implement functional designs including improved social communication, collaborative governance, and urban resilience for the epidemic emergency.” (see Line 447-458 for details)
This manuscript is a resubmission of an earlier submission. The following is a list of the peer review reports and author responses from that submission.
Round 1
Reviewer 1 Report
Some specific issues:
Line 202: how those 83 citied used in the study are selected?
Line 225-227: based on the definition of urban population attraction, the dependent var should have a value around 1, i.e., some cities have a ratio more than and some of them less than 1. However, in table 1, it changes from 55 to 89.36. That is, the definition is not clear enough.
Line 239-240: cities are divided into 5 tiers, but in table 1, urban hierarchy only changes from 1 to 4. Then, it should be treated as a categorical var instead of a continuous one. Moreover, the text should indicate that the meaning of this classification is actually that the lower the number, the larger the size of the city. Therefore, the negative relationships between the urban hierarchy and urban digitalization level and urban pop attraction indicate positive relationships between the size of city and the other two variables.
Table 3: when the coefficient of urban digital transformation level is 0.25 with a SD of 0.52, the corresponding Z value should be 0.5 which is much less than the critical value of 1.96. That is, it is insignificant. (My guess is that the SD could be 0.052)
General comments:
- Cities in China have been affected by covid-19 at quite different levels. Without controlling the level of covid-19 infection, such as the proportion of infected cases, number of deaths of covid-19 and so on, it is meaningless to examine the effect of digital transformation on the recovery of the infected cities.
- One fatal error in the model is the collinearity between the independent and control variable (Pearson’s r = -0.74), which may result in the insignificance of the independent variable.
Author Response
Dear reviewers:
Thank you very much for your letter and for the comments on our manuscript entitled “Can digital transformation promote the rapid recovery of cities from the COVID-19 epidemic? An empirical analysis from Chinese cities” (Submission ID: ijerph-1566936). Those comments are not only valuable and helpful for revising and improving this paper, but also a priceless reference to our future researches. We have studied comments carefully and accordingly made corrections that we hope could meet with approval standards. To help the readers better understand this study, the language of the manuscript has been sent to be edited by the MDPI Author Services (https://www.mdpi.com/authors/english). The main corrections in the paper and the responds to the reviewer’s comments are as following:
Some specific issues:
Point 1: Line 202: how those 83 cities used in the study are selected?
Response 1: We have explained sample selection further in "3.1. Study design and sample selection" to clearly articulate the reasons for our selection of 83 cities. The details are as follows.
“The data are cross-sectional data released by authoritative organizations. Based on the representativeness of cities and the availability of observations of related variables, taking the intersection of the cities covered by the “Research Report on China’s Urban Vitality in the Second Quarter of 2020” released by Baidu and the “2020 China Top 100 Digital Cities Research White Paper” released by CCID Consulting Co. Ltd as the research objects. At last, 83 cities in China were included in the research sample. The sample cities are major cities in China, covering most provinces, as shown in Figure 1.” (see Line 216-223 for details)
Point 2: Line 225-227: based on the definition of urban population attraction, the dependent var should have a value around 1, i.e., some cities have a ratio more than and some of them less than 1. However, in table 1, it changes from 55 to 89.36. That is, the definition is not clear enough.
Response 2: Thanks for pointing out this problem. We are very sorry that we have made an error here. After careful inspection, we found that this error occurred because we placed the descriptive statistics for the dependent variable - "urban population attraction" and the independent variable - "urban digitalization level" in reverse in table 1. In the latest version, we have fixed these bugs. To avoid this error, we recalculated all data and carefully checked the relevant content of the article to ensure that it is accurate. The revised descriptive statistics are as follows.
“Table 1 shows the characteristics of the urban population attraction, the urban digitalization level, and the urban hierarchy of the sample cities. Specifically, the urban population attraction ranges from 0.43 to 16.89, with an average of 3.46 and an SD of 3.58, indicating that the urban population attraction has a large gap between the sample cities, and the overall level is low. The urban digitalization level ranges from 55.02 to 89.36. The average is 69.26, and the SD value is 8.04. This finding shows that although there are gaps in the level of digitization between cities in China, these gaps are smaller than those in the level of population attraction. Additionally, the overall digitalization level of Chinese cities is at a relatively mature stage. This also explains why China has had the ability to deploy digital interventions on a large scale during the COVID-19 outbreak. The urban hierarchy ranges from 1 to 4. That is, all sample cities are Tier 1 to Tier 4 cities. The average is 2.22, and the SD is 0.84, indicating that the sample cities are evenly distributed among different hierarchies. Tier 5 cities were not included in the sample because the cities selected for this study are major cities—large- and medium-sized cities from China. The Tier 5 cities are generally small sized cities, so they were not included in the sample.”(see Line 321-343 for details)
Table 1. Descriptive statistics of the variables.
Variables |
n |
Mean (SD) |
Min |
Max |
Urban population attraction |
83 |
3.46 (3.58) |
0.43 |
16.87 |
Urban digitalization level |
83 |
69.26 (8.04) |
55.02 |
89.36 |
Prevalence of infection |
83 |
0.66 (4.92) |
0.02 |
44.90 |
Case fatality ratio (%) |
83 |
0.94 (1.71) |
0 |
7.69 |
Urban hierarchy |
83 |
2.22 (0.84) |
1 |
4 |
In addition, to better explain why the variable "Urban population attraction" is between 0.43 and 16.89, we have further explained this variable in the "3.3. Dependent variable" section:
“Urban population attraction is the core manifestation of urban vitality. It is calculated by the ratio of the new inflow of a normal population to a specific city to the average new inflow of a normal population to all cities in China. ” (see Line 253-256 for details)
Point 3: Line 239-240: cities are divided into 5 tiers, but in table 1, urban hierarchy only changes from 1 to 4. Then, it should be treated as a categorical var instead of a continuous one. Moreover, the text should indicate that the meaning of this classification is actually that the lower the number, the larger the size of the city. Therefore, the negative relationships between the urban hierarchy and urban digitalization level and urban pop attraction indicate positive relationships between the size of city and the other two variables.
Response 3: Thanks for pointing out this issue, and we have revised the manuscript in response to these comments.
First, Tier 5 cities were not included in the sample cities for the following reasons:
“Tier 5 cities were not included in the sample because the cities selected for this study are major cities—large- and medium-sized cities from China. The Tier 5 cities are generally small sized cities, so they were not included in the sample.” (see Line 341-343 for details)
Second, the relationship between urban hierarchy and urban comprehensive economic and social development level has been explained in the section "3.4 Control variable":
“They are, respectively, valued as: first-tier cities = 1, second-tier cities = 2, third-tier cities = 3, fourth-tier cities = 4, and fifth-tier cities = 5. This means that the lower the urban hierarchy value, the higher the urban comprehensive economic and social development level.” (see Line 284-287 for details)
Finally, since the urban hierarchy is a categorical var, we set the urban hierarchy as a dummy variable (base 1) in the OLS model. The modified OLS results are as follows. (see Line 380-381, table 3 for details)
Table 3. Analysis of the effect of the urban digitalization level on urban population attraction.
Variables |
Coef. (β) |
Std. Err. |
95% Confidence Interval |
Std. Coef. |
t |
P |
vif |
Independent variable |
|
|
|
|
|
|
|
Urban digitalization level |
0.21 |
0.05 |
[0.11, 0.32] |
0.48 |
4.08 |
<0.001 |
3.35 |
Control variables |
|
|
|
|
|
|
|
Prevalence of infection |
0.02 |
0.05 |
[-0.09, 0.12] |
0.02 |
0.33 |
0.740 |
1.29 |
Case fatality ratio |
-0.12 |
0.15 |
[-0.42, 0.18] |
-0.06 |
-0.78 |
0.435 |
1.32 |
Urban hierarchy (base 1) |
|
|
|
|
|
|
|
2 |
-3.12 |
0.81 |
[-4.72, -1.51] |
-0.42 |
-3.86 |
<0.001 |
2.91 |
3 |
-3.62 |
1.08 |
[-5.77, -1.48] |
-0.49 |
-3.37 |
<0.001 |
5.26 |
4 |
-3.27 |
1.69 |
[-6.63, 0.09] |
-0.17 |
-1.94 |
0.057 |
1.92 |
Constant |
-8.54 |
4.19 |
[-16.90, -0.19] |
- |
-2.04 |
0.045 |
- |
Point 4: Table 3: when the coefficient of urban digital transformation level is 0.25 with a SD of 0.52, the corresponding Z value should be 0.5 which is much less than the critical value of 1.96. That is, it is insignificant. (My guess is that the SD could be 0.052)
Response 4: Thank you for pointing out these potentially misleading issues in the manuscript. In the original manuscript, SD is not the Z value or the t value, but the standard error of the independent variable coefficient. To avoid ambiguity, we have changed SD in Table 3 to Sta. Err.. In addition, we added the t value in the OLS model in Table 3, and the independent variable t = 4.08 > 1.96, and P < 0.001. (see Line 380-381, table 3 for details)
Table 3. Analysis of the effect of the urban digitalization level on urban population attraction.
Variables |
Coef. (β) |
Std. Err. |
95% Confidence Interval |
Std. Coef. |
t |
P |
vif |
Independent variable |
|
|
|
|
|
|
|
Urban digitalization level |
0.21 |
0.05 |
[0.11, 0.32] |
0.48 |
4.08 |
<0.001 |
3.35 |
Control variables |
|
|
|
|
|
|
|
Prevalence of infection |
0.02 |
0.05 |
[-0.09, 0.12] |
0.02 |
0.33 |
0.740 |
1.29 |
Case fatality ratio |
-0.12 |
0.15 |
[-0.42, 0.18] |
-0.06 |
-0.78 |
0.435 |
1.32 |
Urban hierarchy (base 1) |
|
|
|
|
|
|
|
2 |
-3.12 |
0.81 |
[-4.72, -1.51] |
-0.42 |
-3.86 |
<0.001 |
2.91 |
3 |
-3.62 |
1.08 |
[-5.77, -1.48] |
-0.49 |
-3.37 |
<0.001 |
5.26 |
4 |
-3.27 |
1.69 |
[-6.63, 0.09] |
-0.17 |
-1.94 |
0.057 |
1.92 |
Constant |
-8.54 |
4.19 |
[-16.90, -0.19] |
- |
-2.04 |
0.045 |
- |
General comments:
Point 5: Cities in China have been affected by covid-19 at quite different levels. Without controlling the level of covid-19 infection, such as the proportion of infected cases, number of deaths of covid-19 and so on, it is meaningless to examine the effect of digital transformation on the recovery of the infected cities.
Response 5: Thanks for pointing out this issue. We have realized that in OLS models, especially in the fields of public health and social science, control variables are always difficult to fully reflect reality due to the availability of data. Although the control variable - urban hierarchy was added in the previous version to comprehensively reflect the business, transportation, population, life, and development potential level of a city, we still feel that this is not enough to reflect the special epidemic emergency. Therefore, we have included in the control variables reflecting the level of covid-19 infection - the prevalence of infection and the case fatality ratio of the COVID-19 epidemic according to your suggestion. The specific modifications are as follows:
“Cities in China have been affected by the COVID-19 epidemic to different extents, leading to differences in the scope and extent of epidemic prevention and control measures. This may also lead to different reactions from the government, society, and the public in a state of crisis. Therefore, the prevalence of infection and the case fatality ratio of the COVID-19 epidemic were included as the control variable of the model as the degree of impact of the epidemic in the sample cities. According to the conventional calculation method of medical statistics, the prevalence of infection of the COVID-19 epidemic in this study refers to the ratio of the confirmed COVID-19 cases of in the entire population sample as of 31 May 2020. It was expressed in this study as the number of confirmed cases per 10,000 population. The case fatality ratio of COVID-19 epidemic represents the ratio of the confirmed cases who died from COVID-19 to the total number of confirmed cases as of 31 May 2020. The number of confirmed COVID-19 cases and deaths were based on data published daily on the official homepages of China's central and provincial health administrations. The number of entire population sample is replaced by the number of permanent residents in official statistics, which comes from the 2020 Provincial Statistical Yearbook released by China.” (see Line 292-308 for details)
“Although the prevalence of infection and the case fatality ratio of the COVID-19 epidemic were not detected to be correlated with independent or dependent variables in this study, we still included them as control variables in the model. This is because these two indicators are widely considered to be important factors in determining the entire epidemic prevention and control process. In addition, the variance inflation factor (vif) values of the independent variable and control variables were within the traditional acceptable range of less than 10, and most of them satisfied the more stringent acceptable threshold of 5. Therefore, the OLS model used in this study can be considered free of multicollinearity.” (see Line 363-371 for details)
Point 6: One fatal error in the model is the collinearity between the independent and control variable (Pearson’s r = -0.74), which may result in the insignificance of the independent variable.
Response 6: Thanks for reminding us about the collinearity between the independent and control variable. We have added a collinearity test to the new version of the OLS model, and the test results show that all variables in the OLS model do not have collinearity within an acceptable threshold. The specific modifications are as follows.
“In addition, the variance inflation factor (vif) values of the independent variable and control variables were within the traditional acceptable range of less than 10, and most of them satisfied the more stringent acceptable threshold of 5. Therefore, the OLS model used in this study can be considered free of multicollinearity.” (see Line 367-371 for details)
Reviewer 2 Report
The authors undertook an interesting research problem concerning the impact of digital transformation on the ability of cities to quickly recover from the COVID-19 epidemic. The theoretical part of the analyses does not raise any objections in principle, the cited literature is up-to-date, adapted in terms of content to the research problem. The authors carried out the empirical study on the example of 83 large- and medium-sized cities in China. The correlation analysis tools used (Pearson correlation analysis and OLS) are correct, although not very advanced, which is largely due to the characteristics of the data obtained. Some doubts can be raised with regard to reducing the assessment of the degree of city recovery after the COVID-19 epidemic only to the assessment of the attractiveness of the urban population in the second quarter of 2020, but the authors are aware of the problems related to the measurement of the degree of city recovery, which they express in the methodological part of the publication. The introduction of a control variable urban hierarchy (including urban socioeconomical characteristics) to the model should be assessed positively, however, it should be taken into account that a variable constructed in that way has limited informative value. It should also be noted that the measurement of the level of digitalisation concerns the end of 2019, and therefore does not take into account potential changes that may have occurred in cities during the first phase of the COVID 19 pandemic. Despite the limitations indicated above, the article is an important voice in the scientific discussion on the impact of digitalisation on the development of Chinese cities during the COVID 19 pandemic.
Author Response
Dear reviewers:
Thank you very much for your letter and for the comments on our manuscript entitled “Can digital transformation promote the rapid recovery of cities from the COVID-19 epidemic? An empirical analysis from Chinese cities” (Submission ID: ijerph-1566936). Those comments are not only valuable and helpful for revising and improving this paper, but also a priceless reference to our future researches. We have studied comments carefully and accordingly made corrections that we hope could meet with approval standards. To help the readers better understand this study, the language of the manuscript has been sent to be edited by the MDPI Author Services (https://www.mdpi.com/authors/english). The main corrections in the paper and the responds to the reviewer’s comments are as following:
Point 1: The authors undertook an interesting research problem concerning the impact of digital transformation on the ability of cities to quickly recover from the COVID-19 epidemic. The theoretical part of the analyses does not raise any objections in principle, the cited literature is up-to-date, adapted in terms of content to the research problem. The authors carried out the empirical study on the example of 83 large- and medium-sized cities in China. The correlation analysis tools used (Pearson correlation analysis and OLS) are correct, although not very advanced, which is largely due to the characteristics of the data obtained. Some doubts can be raised with regard to reducing the assessment of the degree of city recovery after the COVID-19 epidemic only to the assessment of the attractiveness of the urban population in the second quarter of 2020, but the authors are aware of the problems related to the measurement of the degree of city recovery, which they express in the methodological part of the publication. The introduction of a control variable urban hierarchy (including urban socioeconomical characteristics) to the model should be assessed positively, however, it should be taken into account that a variable constructed in that way has limited informative value. It should also be noted that the measurement of the level of digitalisation concerns the end of 2019, and therefore does not take into account potential changes that may have occurred in cities during the first phase of the COVID 19 pandemic. Despite the limitations indicated above, the article is an important voice in the scientific discussion on the impact of digitalisation on the development of Chinese cities during the COVID 19 pandemic.
Response 1: Thank you for your detailed review of our manuscript. Your comments are of great importance to the improvement of the manuscript. And we have carefully read your comments and modified them as much as possible. We realized that there are some contents that need to be improved in the last version according to your comments, especially in terms of control variables and time effects. In this version, we have made the following modifications accordingly.
Regarding control variables, we added two control variables to the OLS model reflecting the level of covid-19 infection - the prevalence of infection and the case fatality ratio of the COVID-19 epidemic. The specific modifications are as follows:
“Cities in China have been affected by the COVID-19 epidemic to different extents, leading to differences in the scope and extent of epidemic prevention and control measures. This may also lead to different reactions from the government, society, and the public in a state of crisis. Therefore, the prevalence of infection and the case fatality ratio of the COVID-19 epidemic were included as the control variable of the model as the degree of impact of the epidemic in the sample cities. According to the conventional calculation method of medical statistics, the prevalence of infection of the COVID-19 epidemic in this study refers to the ratio of the confirmed COVID-19 cases of in the entire population sample as of 31 May 2020. It was expressed in this study as the number of confirmed cases per 10,000 population. The case fatality ratio of COVID-19 epidemic represents the ratio of the confirmed cases who died from COVID-19 to the total number of confirmed cases as of 31 May 2020. The number of confirmed COVID-19 cases and deaths were based on data published daily on the official homepages of China’s central and provincial health administrations. The number of entire population sample is replaced by the number of permanent residents in official statistics, which comes from the 2020 Provincial Statistical Yearbook released by China.” (see Line 292-308 for details)
Regarding the time effects, although we have realized that independent variables and dependent variables may change with time, due to the limitations of cross-sectional data, we can only use the most representative data and the shortest time difference between the variables for analysis. The modifications are as follows:
“In addition, although incorporating the lagged terms of the dependent variable into the model has the potential to remove the effects of more confounding factors in the general case, we did not consider the lagged terms of the dependent variable mainly due to China’s lockdown measures in Wuhan City on 23 January 2020 and travel restrictions on a large scale. This period was the eve of Chinese New Year, the Spring Festival. The Spring Festival is the most important festival in China, and most of the inter-regional workers return to their hometowns to celebrate before the festival, but traveled very little with the outbreak of the COVID-19 epidemic. Therefore, once regional lockdowns and nationwide travel restrictions were lifted, cities that had recovered more from the epidemic were initially more likely to attract more workers. We believe that in this special period, the urban population attraction in the second quarter of 2020 can be well used as an indicator of the degree of recovery form the COVID-19 epidemic in China, and its lag effect does not need to be considered.” (see Line 265-277 for details)
“First, due to the limitations of the data sources, the measurement period of the city's digitization level in this study was in years, while the measurement period of urban population attractiveness was in quarters, which may blur the relationship between the two to a certain extent.” (see Line 438-441 for details)
Reviewer 3 Report
In abstract, I dont know what beta means, until I read the paper. I would eliminate discussion of beta in the paper, because the units of measurement of digitalization are arbitrary. Far better would be to emphasize the share of the variance of the dependent variable that is explained by digitiliazation.
I dont see how the last sentence in the abstract (which is also replicated in the paper) follows from the analysis. I would either tie it in or drop it. I think your point should be that digitalization is an effective way of fighting the virus and by digitizing, other less appealing ways of fighting the virus (e.g. lockdowns and privacy reductions) could be reduced.
you mention that the contribution of two of the authors is the same. Why have the other authors been left out from this note? Perhaps the shares of all the authors should be discussed here.
I would like more discussion of what the variables in section 3.2 mean. I am having trouble visualizing them.
I dont think attraction is your instrumental variable. I think it is your measure of response effectiveness. Please look up the definition of instrumental variable and be sure your statement is accurate.
Nice map.
I believe there is another regression that is simpler and perhaps better. I suspect that highly digitalized cities attract more immigration. which biases your result in favor of the conclusion that digitalization facilitates virus fighting. i suggest a regression of attraction minus attraction lagged as the dependent variable and regress that on digitalization. I encourage the authors to present that regression as well as the regression they have used in the rewrite of the paper. If they chose not to do this, I would like them to explain why they are unable to perform that regression.
Author Response
Dear reviewers:
Thank you very much for your letter and for the comments on our manuscript entitled “Can digital transformation promote the rapid recovery of cities from the COVID-19 epidemic? An empirical analysis from Chinese cities” (Submission ID: ijerph-1566936). Those comments are not only valuable and helpful for revising and improving this paper, but also a priceless reference to our future researches. We have studied comments carefully and accordingly made corrections that we hope could meet with approval standards. To help the readers better understand this study, the language of the manuscript has been sent to be edited by the MDPI Author Services (https://www.mdpi.com/authors/english). The main corrections in the paper and the responds to the reviewer’s comments are as following:
Point 1: In abstract, I don’t know what beta means, until I read the paper. I would eliminate discussion of beta in the paper, because the units of measurement of digitalization are arbitrary. Far better would be to emphasize the share of the variance of the dependent variable that is explained by digitiliazation.
Response 1: Thank you for pointing out this problem, and it made us realize that the Beta(β)and R2 are not required in the Abstract part and may cause ambiguity without any explanation. Therefore, we removed Beta (β) and R2 in the Abstract section based on your comments. And we made sure that this change didn't affect what the Abstract section needed to express. At the same time, in order to eliminate the influence of the units of different variables, we added a standardized coefficient (Std.Coef.) in Table 3, so that the magnitude of the coefficient can reflect the effect of the urban digitalization level on the urban population attraction. And it can be compared with the size of the effect of the control variable. (see Line 380-381, table 3 for details)
Table 3. Analysis of the effect of the urban digitalization level on urban population attraction.
Variables |
Coef. (β) |
Std. Err. |
95% Confidence Interval |
Std. Coef. |
t |
P |
vif |
Independent variable |
|
|
|
|
|
|
|
Urban digitalization level |
0.21 |
0.05 |
[0.11, 0.32] |
0.48 |
4.08 |
<0.001 |
3.35 |
Control variables |
|
|
|
|
|
|
|
Prevalence of infection |
0.02 |
0.05 |
[-0.09, 0.12] |
0.02 |
0.33 |
0.740 |
1.29 |
Case fatality ratio |
-0.12 |
0.15 |
[-0.42, 0.18] |
-0.06 |
-0.78 |
0.435 |
1.32 |
Urban hierarchy (base 1) |
|
|
|
|
|
|
|
2 |
-3.12 |
0.81 |
[-4.72, -1.51] |
-0.42 |
-3.86 |
<0.001 |
2.91 |
3 |
-3.62 |
1.08 |
[-5.77, -1.48] |
-0.49 |
-3.37 |
<0.001 |
5.26 |
4 |
-3.27 |
1.69 |
[-6.63, 0.09] |
-0.17 |
-1.94 |
0.057 |
1.92 |
Constant |
-8.54 |
4.19 |
[-16.90, -0.19] |
- |
-2.04 |
0.045 |
- |
Point 2: I don’t see how the last sentence in the abstract (which is also replicated in the paper) follows from the analysis. I would either tie it in or drop it. I think your point should be that digitalization is an effective way of fighting the virus and by digitizing, other less appealing ways of fighting the virus (e.g. lockdowns and privacy reductions) could be reduced.
Response 2: Thank you very much for your valuable suggestion, and we have removed the last sentence of the abstract. Some content has been revised to ensure a more accurate summary of the full text. The modified contents are as follows.
“Conclusions: A positive relationship was found between urban digital transformation and the rapid recovery of cities from the COVID-19 epidemic. Digital inventions for social communication, collaborative governance, and urban resilience is an effective way of fighting the COVID-19 emergency.” (see Line 41-48 for details)
Point 3: You mention that the contribution of two of the authors is the same. Why have the other authors been left out from this note? Perhaps the shares of all the authors should be discussed here.
Response 3: Thank you very much for these valuable comments, and we realize that clarifying the authors' contribution is extremely important to respect the authors' contribution, so we elaborate on the division of labor and contribution of each author in the "Author Contributions" section. Nonetheless, the time and effort each author devotes to this manuscript varies, and it is generally accepted that the first author contributes the greatest effort throughout the entire process of writing the original draft. However, in this manuscript, the first two authors are considered by us to have made the same effort in writing the original draft, so the manuscript emphasizes the contribution of two of the authors is the same, while the last two authors are listed in the usual way out. The contributions of each author are described below.
“Jiaojiao Liu: Visualization, methodology, writing—original draft. Shuai Liu: Data curation, writing—original draft. Xiaolin Xu: Conceptualization, writing—reviewing and editing. Qi Zou: Conceptualization, methodology, writing—original draft, supervision. All authors revised the manuscript.” (see Line 474-478 for details)
Point 4: I would like more discussion of what the variables in section 3.2 mean. I am having trouble visualizing them.
Response 4: Thank you very much for your valuable suggestion, and we've covered the independent variable-urban digitalization level in the "3.2. Independent variable" section. The additions are as follows.
“The independent variable is the urban digitalization level. Urban digitization means that cities use ICT technology to build various information infrastructures and application capabilities, coupled with urban scene resources, and realize the integration and symbiosis of the “digital world” and the “physical city” [34]. Urban digitalization is manifested in various aspects such as industrial development, urban construction, government management, social governance, and public services. ” (see Line 230-235 for details)
Point 5: I don’t think attraction is your instrumental variable. I think it is your measure of response effectiveness. Please look up the definition of instrumental variable and be sure your statement is accurate.
Response 5: Thank you very much for your valuable suggestion, we have removed the elaboration on "instrumental variables" to avoid semantic confusion in the metrological approach. The relevant content is modified as follows.
“We considered introducing urban population attraction in the second quarter of 2020 as an indicator to appropriately measure the recovery of cities from the COVID-19 epidemic.” (see Line 250-252 for details)
Point 6: I believe there is another regression that is simpler and perhaps better. I suspect that highly digitalized cities attract more immigration. which biases your result in favor of the conclusion that digitalization facilitates virus fighting. I suggest a regression of attraction minus attraction lagged as the dependent variable and regress that on digitalization. I encourage the authors to present that regression as well as the regression they have used in the rewrite of the paper. If they chose not to do this, I would like them to explain why they are unable to perform that regression.
Response 6: Thank you very much for your valuable suggestion. We realized that lagged terms of the dependent variable could be included in the OLS model to remove the influence of possible time effects. However, due to the particularity of the time selected in this study, we believe that the time points corresponding to the selected variables can greatly reduce the influence of time effects, so there is no need to incorporate the lagged terms of the dependent variable into the model. In addition, the limitations of cross-sectional data made it difficult for us to make this attempt, so we included this matter in the "5.2. Limitations" section. The specific reasons and modifications are as follows.
“In addition, although incorporating the lagged terms of the dependent variable into the model has the potential to remove the effects of more confounding factors in the general case, we did not consider the lagged terms of the dependent variable mainly due to China’s lockdown measures in Wuhan City on 23 January 2020 and travel restrictions on a large scale. This period was the eve of Chinese New Year, the Spring Festival. The Spring Festival is the most important festival in China, and most of the inter-regional workers return to their hometowns to celebrate before the festival, but traveled very little with the outbreak of the COVID-19 epidemic. Therefore, once regional lockdowns and nationwide travel restrictions were lifted, cities that had recovered more from the epidemic were initially more likely to attract more workers. We believe that in this special period, the urban population attraction in the second quarter of 2020 can be well used as an indicator of the degree of recovery form the COVID-19 epidemic in China, and its lag effect does not need to be considered.” (see Line 265-277 for details)
“First, due to the limitations of the data sources, the measurement period of the city's digitization level in this study was in years, while the measurement period of urban population attractiveness was in quarters, which may blur the relationship between the two to a certain extent.” (see Line 438-441 for details)
Round 2
Reviewer 1 Report
Line 214-218: it seems to me that those 83 cities used in the study are those in the “2020 China Top 100 Digital Cities Research White Paper” with available “related variables.” If that is the case, these cities are not a random sample of all cities in China. Then, the representativeness of these cities cannot be guaranteed and the significance test should be out of question. For example, small cities (tier 5) have been completely excluded from the study.
Line 248-250: my understanding of the definition of the dep variable, urban population attraction, is the # of inflow population divided by the average inflow population per city. This measure is strongly correlated with the population size of the city, Urban hierarchy (the Pearson’s r is -0.74 in table 2), i.e., the larger the city, the more the floating population. It is also strongly associated with digitalization level (r = 0.79). However, this measurement of the recovery from covid-19 is not associated with the prevalence of covid-19 and fatality ratio of covid-19 (the correlation coefficients are not significant in table 2 and they do not have significant effect on the dep variable in table 3). That is, it is not a valid measure of the recovery from the pandemics. Maybe, the change of this ratio from before the epidemics to the after epidemics is a good measure, which should be related to the occurrence of covid-19.
Table 2: As urban hierarchy is an ordinal variable, the Pearson’s r cannot be used to measure the association between urban hierarchy and other continuous variables.
Table 3: β is the notation of a standardized coefficient. The Coef. Column in table 3 should be b value, not β. Info in table 3 is redundant.
In sum, the research design has fatal flaws. Firstly, the measure of the dependent variable is not valid. Secondly, the non-random sample used in the study cannot represent all cities in China.
Reviewer 3 Report
Ready to publish. The authors did a nice job.